# Intuitive Multilingual Audio-Visual Speech Recognition with a Single-Trained Model

**Joanna Hong, Se Jin Park, Yong Man Ro**[†]
School of Electrical Engineering, KAIST
{joanna2587, jinny960812, ymro}@kaist.ac.kr

## Abstract

We present a novel approach to multilingual audio-visual speech recognition tasks by introducing a single model on a multilingual dataset. Motivated by a human cognitive system where humans can intuitively distinguish different languages without any conscious effort or guidance, we propose a model that can capture which language is given as an input speech by distinguishing the inherent similarities and differences between languages. To do so, we design a prompt fine-tuning technique into the largely pre-trained audio-visual representation model so that the network can recognize the language class as well as the speech with the corresponding language. Our work contributes to developing robust and efficient multilingual audio-visual speech recognition systems, reducing the need for language-specific models.

## 1 Introduction

With the great advancements of deep learning, automatic audio-visual speech recognition (AVSR) technology has achieved remarkable progress (Mroueh et al., 2015; Afouras et al., 2018a; Baevski et al., 2020; Kim et al., 2022; Ma et al., 2021; Hong et al., 2023). It utilizes multimodal inputs, including both audio and visual cues, providing several advantages to the deep learning-based speech recognition branch. One of the benefits is that it can accurately recognize speech in noisy environments, such as crowded restaurants or corrupted video conferencing situations. This capability is critical for advancing the field of automatic speech recognition technology in the future.

Nevertheless, outstanding performances in audio-visual speech recognition have been mostly shown in monolingual datasets, particularly in English. Few recent studies have started focusing on multilingual speech recognition tasks, but they are still in their infancy. One work (Zinonos et al.,

2023) has presented cross-lingual visual speech representation learning and shows multilingual models with more data outperform monolingual ones. The other works, MuAViC (Anwar et al., 2023) and MixSpeech (Cheng et al., 2023), have newly introduced a multilingual audio-visual corpus for speech recognition and speech-to-text translation task.

While the recent multilingual speech recognition studies have shown remarkable advances, they have only focused on pre-training the multilingual speech recognition model or audio-visual speech representation model, followed by fine-tuning with the specific language (Zinonos et al., 2023). This is due to the imbalance of dataset language distribution and each language's distinctive characteristic. However, producing a language-specific speech recognition model can be time-consuming and inefficient. Most importantly, it does not correspond to a real-life situation, where humans intuitively recognize the language when others are speaking.

Inspired by the human understanding perspective, in this paper, we design a single model multilingual audio-visual speech recognition framework that the model can not only determine which language is taken into the input speech but also recognize the speech correctly. To do so, we newly introduce an audio-visual guiding linguistic representation extractor. With the largely trained audio-visual speech representation model (Shi et al., 2022), we fine-tune the model by utilizing prompts so that the model can extract comprehensive linguistic information from the audio and video inputs. We set only a small amount of downstream task-specific parameters as the learnable parameters for extracting linguistic representation into the input space so that comprehensive linguistic information can be produced. Furthermore, we consider the imbalanced distribution issue that the multilingual datasets contain, with some languages having significantly fewer samples than others. Inspired by (Li et al., 2022), we suggest a weighted objective

---

[†]Corresponding Author.

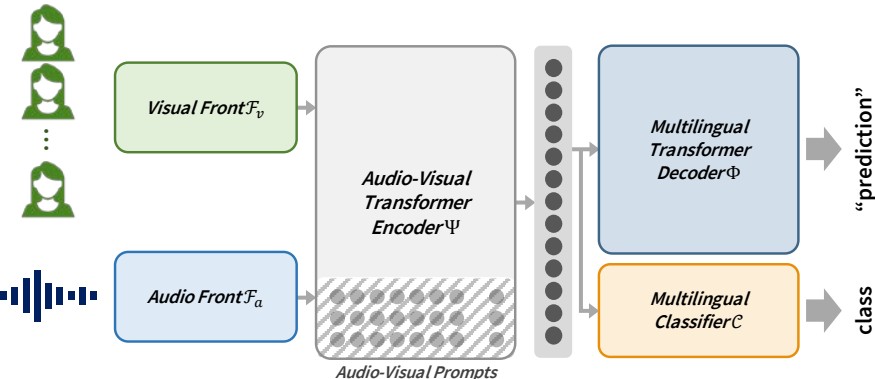

Figure 1: Overall architecture of the proposed multilingual audio-visual speech recognition model.

function in order to balance each language distribution. By training our model on a diverse set of languages, we aim to capture their inherent similarities and differences, allowing our model to recognize and transcribe speech in multiple languages with greater accuracy and efficiency. We validate the effectiveness of our proposed model using MuAViC (Anwar et al., 2023), a multilingual audio-visual corpus for robust speech recognition and robust speech-to-text translation providing 1200 hours of audio-visual speech in 9 languages. Therefore, our work contributes to developing efficient and robust multilingual audio-visual speech recognition systems. We also believe that our proposed approach has several potential benefits: reducing the need for language-specific models, improving the performance of speech recognition in low-resource languages, and enabling more effective multilingual communication.

## 2 Methodology

Given the input multilingual video sequence, $\boldsymbol{x}_v = \{x_1, \dots, x_L\} \in \mathbb{R}^{L \times H \times W \times C}$ where $L$, $H$, $W$, and $C$ are the frame length, height, width, and channel sizes, respectively, and the paired input audio sequence, $\boldsymbol{x}_a \in \mathbb{R}^S$, where $S$ represents the length of audio, we design a model that properly recognizes the given input video and audio with the correct language. We aim to utilize both visual and audio information so that the proposed architecture can successfully recognize not only the language but also the content of the input video. To this end, we initially propose a language prompt adopted from the largely pre-trained audio-visual speech representation model (Shi et al., 2022), which we call it linguistic representation extractor. Further, we design a multilingual transformer decoder given the inputs of language class, linguistic representation, and the combined features from the audio and

visual transformer encoders. We will explain the detailed aforementioned techniques in the following subsections.

### 2.1 Linguistic Representation Extractor

The first thing when recognizing one's speech in a multinational society is to distinguish the language the speaker is presenting. When identifying the language of the speech, it is important to verify the speaker's accent, pronunciation, and representative vocabulary from the speaker's facial movements and audio information. Inspired by the intuitive human understanding procedure, we design a linguistic representation extractor from the largely pre-trained audio-visual speech representation model. We add trainable continuous embeddings, so-called prompts, to the original sequence of input features in order to fine-tune the pre-trained model relevant to recognizing multilingual input signals.

### 2.1.1 Prompt fine-tuning for Linguistic Representation Extractor

For the input multilingual video sequence $\boldsymbol{x}_v$ and audio sequence $\boldsymbol{x}_a$, audio features $f_a \in \mathbb{R}^{T \times D}$ and visual features $f_v \in \mathbb{R}^{T \times D}$, are extracted from through Audio Front and Visual Front, respectively.

$$f_v = \mathcal{F}_v(\boldsymbol{x}_v) \quad \text{and} \quad f_a = \mathcal{F}_a(\boldsymbol{x}_a). \quad (1)$$

The audio features $f_a$ and the visual features $f_v$ are concatenated followed by layer normalization and linear projection, producing the audio-visual features $f_{av} \in \mathbb{R}^{T \times D}$. Then, we apply Audio-Visual prompts into every layer of the Audio-Visual Transformer Encoder $\Psi$, along with the audio-visual features $f_{av}$. For each audio-visual prompt, we assign language-representative prompts, $P \in \mathbb{R}^{n \times d}$, which are trained to extract linguistic-relative features from the pre-trained audio-visual representa-

tion model. Here, $n$ is the number of prompts.

$$(\_, f_{av,i}) = \Psi_i(P_{av,i-1}, f_{av,i-1}) \qquad (2)$$

$$(\hat{P}_{av,L}, f_{av,L}) = \Psi_L(P_{av,L-1}, f_{L-1}), \qquad (3)$$

for the $i$-th layer $\Psi_i$, where $i = 1, 2, \ldots, L$. We now call $\mathbf{e}_{av} = (\hat{P}_{av,L}, f_{av,L}) \in \mathbb{R}^{(n+T) \times d}$, audio-visual prompt embedding feature.

### 2.1.2 Multilingual Classifier for Linguistic Class Prompt

After extracting the audio-visual prompt embedding feature $\mathbf{e}_{av}$, we firstly train the prompt tuning module by updating the gradient of the learnable parameters of the prompts with the classification loss. As indicated in Figure 1, we propose a multilingual classifier in order to distinguish the input language class. The multilingual classifier $\mathcal{C}$ consists of four blocks of 1D convolution layer, batch normalization, and Relu activation, followed by two linear layers with Relu activation.

$$logit_{pred} = \mathcal{C}(\mathbf{e}_{av}). \qquad (4)$$

The output language class is $logit_{pred} \in \mathbb{R}^m$, where $m$ represents the number of languages for training. Then, the logit is updated through cross-entropy objective function:

$$\mathcal{L}_{class} = CE(logit_{pred}, logit_{gt}). \qquad (5)$$

Therefore, by updating the correct language label, the model can be provided with guidance when recognizing the multilingual speech correctly in the backbone network that will be expressed in further sections.

### 2.2 Objective Functions

The proposed multilingual AVSR framework is trained in an end-to-end manner. For the objective function, we utilize joint CTC/attention (Kim et al., 2017). CTC (Graves et al., 2006) loss is defined as:

$$p_c(y|x) \approx \Pi_{t=1}^{T} p(y_t|x), \qquad (6)$$

with an independent assumption of each output, and attention-based loss is defined as:

$$p_a(y|x) = \Pi_{j=1}^{N} p(y_j|y_{<j}, x). \qquad (7)$$

Here, the current prediction is determined by previous predictions and inputs, thus including the learning of the internal language model, where

$N$ represents the total length of ground-truth text. Then, the total objective can be written as follows,

$$\mathcal{L}_{ctc} = \log p_a(y|x), \qquad (8)$$

$$\mathcal{L}_{att} = \log p_c(y|x), \qquad (9)$$

$$\mathcal{L}_{total} = \alpha \cdot \mathcal{L}_{ctc} + (1 - \alpha) \cdot \mathcal{L}_{att} + \beta \cdot \mathcal{L}_{class}, \qquad (10)$$

where $\alpha$ and $\beta$ are weight parameters for balancing three loss terms.

### 2.2.1 Objective Functions for Balancing the Language Distribution

An objective function weight is designed to balance the distribution of language data, due to the issue of language imbalance in the multilingual dataset. This weight $\gamma$ is calculated as the inverse root of the data distribution ratio $r$ for each language in each mini-batch:

$$\gamma = \frac{1}{\sqrt{r}}. \qquad (11)$$

Therefore, the updated total objective function can be re-written as follows,

$$\mathcal{L}_{total} = \gamma \cdot (\alpha \cdot \mathcal{L}_{ctc} + (1 - \alpha) \cdot \mathcal{L}_{att} + \beta \cdot \mathcal{L}_{class}). \qquad (12)$$

The rationale behind this design comes from the observation that the multilingual dataset often exhibits an uneven distribution of samples across different languages. Thus, when updating the loss during training, it becomes crucial to employ a balancing loss function with a smaller weight for languages that contain a larger number of samples and a larger weight for languages that have fewer samples.

By incorporating this weight into the objective function, the model is encouraged to assign greater importance to underrepresented languages during the learning process. This approach aims to mitigate the adverse effects of language imbalance and prevent the model from being biased toward dominant languages. Thus, the model becomes more capable of effectively recognizing and transcribing speech in languages with limited available data.

## 3 Experimental Setup

### 3.1 Datasets

The MuAViC dataset (Anwar et al., 2023) is a multilingual audio-visual corpus consisting of roughly 1,200 hours of transcribed data spanning 9 languages: English, Arabic, German, Greek, Spanish, French, Italian, Portuguese and Russian. It is collected from TED and TEDx talk recordings, where

| Type | Model | Ar | De | El | En | Es | Fr | It | Pt | Ru | Avg |
|------|-------|-----|-----|-----|-----|-----|-----|-----|-----|-----|-----|
| **Clean** | Monolingual [R6] | 99.24 | 53.93 | 25.85 | 2.21 | 16.66 | 26.26 | 20.07 | 19.97 | 32.77 | 33.00 |
| | Multilingual [R6] | 90.49 | 56.01 | 37.96 | – | 18.99 | 22.97 | 21.82 | 22.34 | 45.50 | 39.51 |
| | Proposed Model ($\mathcal{L}_{att}$) | 91.48 | 52.60 | 45.84 | 3.39 | 20.28 | 23.76 | 21.45 | 23.58 | 57.36 | 37.75 |
| | Proposed Model ($\mathcal{L}_{att} + \mathcal{L}_{ctc}$) | 90.94 | 48.10 | 42.41 | 2.47 | 18.04 | 21.52 | 19.67 | 20.80 | 54.12 | 35.34 |
| **Noisy** | Monolingual [R6] | 100.19 | 70.49 | 50.81 | 6.50 | 40.72 | 44.87 | 47.90 | 42.30 | 49.48 | 50.36 |
| | Multilingual [R6] | 98.65 | 74.41 | 62.98 | – | 42.11 | 41.86 | 47.22 | 44.86 | 65.93 | 59.75 |
| | Proposed Model ($\mathcal{L}_{att}$) | 100.017 | 70.56 | 66.43 | 10.68 | 43.48 | 41.87 | 47.98 | 46.73 | 74.00 | 55.75 |
| | Proposed Model ($\mathcal{L}_{att} + \mathcal{L}_{ctc}$) | 99.66 | 65.92 | 61.94 | 9.01 | 38.94 | 37.37 | 42.48 | 42.13 | 68.99 | 51.83 |

Table 1: WER (%) comparisons with the previous multilingual audio-visual speech recognition methods on clean and noisy settings.

native or non-native speakers (only one speaker most of the time) deliver public speech on stage and cameras capture stage scenes switching among different viewpoints.

## 3.2 Implementation Details

We use largely pre-trained visual frontend, audio frontend, and audio-visual transformer encoder (Shi et al., 2022), trained on LRS3-TED (Afouras et al., 2018b) and VoxCeleb2 English (Chung et al., 2018). We fine-tune the encoding models guided by the prompts and the multilingual classifier in an end-to-end manner, such that the prompts learn the meaningful content, *i.e.*, the language class of the input speech and how the speech is delivered, for making the correct prediction in the multilingual scheme. We set $\alpha = 0.1$ and $\beta = 10.0$. For training and testing, we follow the same noise injection protocol as (Anwar et al., 2023).

## 4 Experimental Result

In the experimental result in Table 1, we report the performances of our proposed model with attention loss only and both attention and CTC loss along with the previous multilingual model (Anwar et al., 2023) performance. The monolingual model refers to the model that is separately trained on each language and the multilingual refers to the model that is jointly trained on all the 8 non-English languages. The previous model did not include En in their multilingual model, so it remains blank. Note that we test the provided trained model for reporting previous work performance.

**Clean Environment.** We evaluate audio-visual speech recognition in a clean environment. As shown in the first section of Table 1, our proposed model outperforms the previous model (Anwar et al., 2023) in several languages, where the greatest improvement (7.91 WER reduction, 14% rel-

ative) has been made in one of the low-resourced language, German (DE). Such results demonstrate that the proposed method effectively enables multilingual audio-visual speech recognition in a unified framework, improving the performance in low-resource languages.

**Noisy Environment.** In the second section of Table 1, we evaluate the proposed method in a noisy setup. The proposed model achieves average WER of 47.18, excluding English while the previous multilingual model achieves 59.75 WER, which is a 4.31% relative improvement. The performance gap between the previous monolingual model (Anwar et al., 2023) is even smaller with an average of 2.4% relative WER. We contribute such improvement to the language-representative prompt which allows the model to embed language-specific features from the audio-visual input, simulating the monolingual framework.

## 5 Conclusion

We introduce the multilingual audio-visual speech recognition model by training a single model on a diverse set of languages. The proposed model finetunes a largely trained audio-visual representation model with prompts to provide meaningful language information. It has presented a promising starting point for future research endeavors.

**Acknowledgement** This work was partially supported by the National Research Foundation of Korea (NRF) grant funded by the Korea government (MSIT) (No.NRF-2022R1A2C2005529), and Institute of Information & communications Technology Planning & Evaluation (IITP) grant funded by the Korea government(MSIT) (No.2022-0-00124, Development of Artificial Intelligence Technology for Self-Improving Competency-Aware Learning Capabilities).

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
