# OpenReview forum: "Intuitive Multilingual Audio-Visual Speech Recognition with a Single-Trained Model"
_EMNLP/2023/Conference — EMNLP 2023 Findings_

### Official Review · Reviewer_oy3M · 2023-08-01

**Soundness:** 4

**Excitement:**

4: Strong: This paper deepens the understanding of some phenomenon or lowers the barriers to an existing research direction.

**Paper Topic And Main Contributions:**

This paper presents a novel approach for multilingual audio-visual speech recognition tasks by developing a single model on a multilingual dataset. The primary problem being addressed in the paper is the limited success of current audio-visual speech recognition models in multilingual datasets, as they are mostly focused on monolingual data, particularly English. The paper aims to design a single model capable of effectively distinguishing the language and properly recognizing the input speech in multiple languages.

The main contributions of the paper are as follows:

1. Propose a single model framework for multilingual audio-visual speech recognition, inspired by the intuitive human language understanding process. The model is designed to determine the language of the input speech and recognize the speech correctly.

2. Introduce an audio-visual guiding linguistic representation extractor by fine-tuning a largely pre-trained audio-visual representation model using prompts to provide language information. This allows the model to extract comprehensive linguistic information from the audio and video inputs.

3. Address the issue of imbalanced language distribution in multilingual datasets by employing a weighted objective function that balances the distribution of language data while updating the loss during training. This aims to improve the performance of speech recognition in low-resource languages and prevent the model from being biased toward dominant languages.

4. Validate the effectiveness of the proposed model by conducting experiments on MuAViC, a multilingual audio-visual corpus containing 9 languages. The results show that the proposed model outperforms the previous multilingual model with an average 11% WER reduction in a clean environment and an average 13% WER reduction in a noisy environment.

**Questions For The Authors:**

A. It is not clear in the methodology section how the trainable continuous embeddings (prompts) are generated and optimized during the fine-tuning process. Can such prompts be learned automatically, or do they require manual annotation? Please provide more details on this process to help readers better understand how these prompts contribute to the overall model.

B. In the experiments, the proposed model is compared primarily with the work of Anwar et al. (2023). It would be beneficial to include comparisons with other relevant methods and show how the proposed approach performs in comparison to a broader range of existing methods in the field of multilingual AVSR. Are there other methods that could be included for comparison?

C. The paper focuses on a dataset containing nine languages which may be considered heavily biased towards Indo-European languages or the languages with a relatively large amount of available data. How would the proposed method perform for languages with few data samples or for languages from very different linguistic families (e.g., tonal languages, agglutinative languages)? Are there plans to extend or validate the model on more diverse sets of languages in the future?

D. The model has been tested in both clean and noisy environments. Although it shows improvement over the previous multilingual model, the performance seems to be lower in noisy environments. What are the possible reasons for the reduced performance, and could this be improved upon with further modifications to the model architecture or training strategy?

**Reasons To Accept:**

1. Novelty: The approach of fine-tuning a largely pre-trained audio-visual representation model with language prompts is new and innovative, enabling the model to effectively recognize and transcribe speech in multiple languages.

2. Robustness: The proposed model demonstrates performance improvements in low-resource languages and exhibits strong results in both clean and noisy environments. This has significant practical implications in real-world applications where speech recognition systems often face challenges in noisy conditions.

3. Addressing Language Imbalance: The paper proposes an objective function weight to balance the distribution of language data, mitigating potential biases towards dominant languages while improving the recognition and transcription of low-resource languages.

4. Comprehensive Experiments: The experiments are conducted on a recent and challenging dataset (MuAViC), consisting of 1200 hours of audio-visual speech data in 9 languages, providing a comprehensive evaluation of the model.

5. Potential Impact: The proposed single model approach can reduce the need for language-specific models and improve the performance of speech recognition in low-resource languages. This would benefit the NLP community in developing more efficient and robust multilingual speech recognition systems.

**Reasons To Reject:**

1. One weakness is the lack of an in-depth ablation study on the proposed approach. Some questions remain unanswered, such as the effectiveness of prompt fine-tuning in various settings and the sensitivity of the model to the choice of hyperparameters.
2. The paper does not provide results for a setting where the language is unknown during inference. Dealing with an unknown language is a crucial aspect in real-world scenarios.
3. The authors mention the imbalance in the dataset, but the exact distribution is not provided. It is important to understand how each language is uniformly represented in experiments and the impact on the overall performance.
4. Although the proposed approach performs better than existing models, it may still be interesting to investigate how this method can benefit from other recent advancements such as self-supervised learning or transformers in multimodal learning.
5. The paper primarily focuses on the MuAViC dataset. It would be beneficial to consider other datasets to explore the generalization of the proposed method across diverse data sources.

**Reproducibility:**

3: Could reproduce the results with some difficulty. The settings of parameters are underspecified or subjectively determined; the training/evaluation data are not widely available.

**Reviewer Confidence:**

4: Quite sure. I tried to check the important points carefully. It's unlikely, though conceivable, that I missed something that should affect my ratings.

---

> ### Author Rebuttal · Authors · 2023-08-29
>
> We would like to express our gratitude for your valuable feedback. We hope this rebuttal helps to elucidate any confusing subjects such that reviewers would be open to improve their ratings. We will incorporate all feedback in our final manuscript.
>
> **(1) The lack of an in-depth ablation study**
>
> We would like to thank the reviewer for delivering the concerns about the manuscript. We totally understand the concerns that the reviewer has raised. We would like to carefully mention that through different settings for prompt fine-tuning, we come up with the most comparable setting to the previous model. We will conduct more analysis on model performances using various choices of hyperparameters in the future manuscript.
>
> **(2) Unknown language setting**
>
> We would greatly appreciate the reviewer’s suggestion. We would like to clarify that we mainly contribute to design the singly modeled multilingual audio-visual speech recognition technique with the use of a language classification module and the novel prompt finetuning technique for the first time. In regarding the aspect of real-world scenarios, we totally agree that it is crucial to have the model recognize speech from an unknown language, and this could be the next step to focus on researching the robust multilingual speech recognition model, even in unknown languages.
>
> **(3) Distribution of imbalanced dataset**
>
> To address the concern of the reviewer, we kindly report the duration statistics of muavic dataset for audio-visual speech recognition in hours of the MuAViC dataset. We refer to the original manuscript of the MuAViC (Anwar et al.,2023).  We will add the additional information about the imbalanced distribution of the multilingual dataset in the future manuscript.
>
> |    | Train | Dev | Test |
> |----|-------|-----|------|
> | **En** |  436.0 | 1.1 | 0.8  |
> | **Ar** | 16.0  | 1.5 | 1.2 |
> | **De** | 10.0  | 1.6 | 1.5 |
> | **El** |  25.0 | 2.3 | 2.0 |
> | **Es** | 178.0 | 1.6 | 1.7  |
> | **Fr** |  176.0 | 1.8 | 1.5  |
> | **It** |101.0  | 1.9 | 1.9|
> | **Pt** |  153.0  | 1.5 | 1.8  |
> | **Ru** |  49.0 | 1.7 | 1.8|
>
> **(4) Investigating on other recent advancements**
>
> We would appreciate the valuable suggestion. We clearly understand that comparing with the recent advancements like self-supervised techniques or architectures like transformers in multimodal learning is beneficial for the model analysis. We would like to clarify that the previous architecture and the largely pre-trained audio-visual representation model (Shi et al., 2022) deal with the self-supervised learning technique and utilize transformer.
>
> **(5) Experiments on other datasets**
>
> We thank the reviewer’s concern focusing on the MuAViC dataset. We would like to kindly address that we primarily focus on the dataset because it is the largest open benchmark for multilingual AVSR [1]. The dataset consists of 1200 hours of transcribed AV speech in nine different languages from 8000+ speakers in TED and TEDx talks [2] which has been widely used for multilingual speech recognition. As the reviewer noted, incorporating a more diverse range of data would further enhance the proposed model’s robustness and capacity for generalization.  We will utilize different multilingual datasets for the generalization of the proposed method in the future manuscript.
>
> $\newline$
>
> **Additional Questions**
>
> **A.** __*(Prompt learning)*__ &nbsp; The proposed language-representative prompts do not need any annotation during training. With the end-to-end training of the entire proposed architecture, the prompts are expected to learn the meaningful contents (i.e., which language the input speech contains and how the speech is delivering) for the correct prediction. We will clearly state the methodology section in the future manuscript to avoid any confusion.
>
> **B.** __*(Comparison to other methods)*__ &nbsp; We would like to thank the reviewer for the valuable suggestion. We make a comparison with the previous model that has conducted experiments on multilingual audio-visual speech recognition with a single model without any extra language class information.  There are a few multilingual speech recognition tasks like [3], but it has been conducted with a single input modality. We will add an additional performance comparison with the same setting that we have conducted in the experiment in the future manuscript.
>
> **C.** __*(Experiments on other languages)*__ &nbsp; We would like to kindly point out that to the best of our knowledge, it is the first time to focus on designing a single model for the multilingual audio-visual speech recognition task, so we start from designing the model with the most general languages. Our model has been able to improve the performance in low-resource languages in the training dataset and reduce the need for language-specific models, enabling more effective multilingual communication. Exploring robustness to languages from very different linguistic families would be the subsequent research direction to extend the generalization capacity of the proposed model. We have plans to build upon variances of linguistic families by incorporating AVSpeech [4] dataset which contains 4700 hours of audio-visual paired clips in 95 different languages. We believe that our work has potential benefits in reducing the need for language-specific models, improving the performance of speech recognition in low-resource languages, and enabling more effective multilingual communication.
>
> **D.** __*(Performances in noisy environments)*__ &nbsp; It has always been challenging to recognize the speech from the noisy input, as the baseline architecture (Anwar et al.,2023) has shown degraded performances in the noisy environment. The reduced performance is generally caused by the distraction of noises inserted from the clean audio waveform so that the network can hardly distinguish between the clean audio waveform and the noise. We can firstly enhance the noisy audio features by adapting the feature enhancement module like [5] to create the enhanced audio features and take them into the recognition model.
>
> $\newline$
>
> [1] Ivanko, Denis, Dmitry Ryumin, and Alexey Karpov. "A Review of Recent Advances on Deep Learning Methods for Audio-Visual Speech Recognition." Mathematics 11.12 (2023): 2665.
>
> [2] Salesky, Elizabeth, et al. "The multilingual tedx corpus for speech recognition and translation." arXiv preprint arXiv:2102.01757 (2021).
>
> [3] Zinonos, Andreas, et al. "Learning Cross-Lingual Visual Speech Representations." ICASSP 2023-2023 IEEE International Conference on Acoustics, Speech and Signal Processing (ICASSP). IEEE, 2023.
>
> [4] Ephrat, Ariel, et al. "Looking to listen at the cocktail party: A speaker-independent audio-visual model for speech separation." arXiv preprint arXiv:1804.03619 (2018).
>
> [5] Hong, Joanna, et al. "Visual context-driven audio feature enhancement for robust end-to-end audio-visual speech recognition." arXiv preprint arXiv:2207.06020 (2022).

---

### Official Review · Reviewer_EVgV · 2023-08-04

**Soundness:** 2

**Excitement:**

2: Mediocre: This paper makes marginal contributions (vs non-contemporaneous work), so I would rather not see it in the conference.

**Paper Topic And Main Contributions:**

This paper proposes a single multilingual AVASR model for handling audio+visual input in multiple languages.  The model hypothesizes detected language in addition to transcript for spoken input.  It utilizes an encoder decoder architecture where the encoder is a pre-trained AV transformer model, and decoder is a multi-lingual transformer architecture.  A classifier is used for predicting the language of input speech.  Proposed model is trained/evaluated on the recently released MuAViC dataset.

**Questions For The Authors:**

Please identify which results from Anwar et al. are cited in the paper.

**Reasons To Accept:**

* Formulation and empirical study of a single simple multi-lingual AVASR model


**Reasons To Reject:**

* In the results presented in Table 1, the baseline numbers from Anwar et al don’t match those reported in Table 1.  Not sure which paper to look at for source of the reported Anwar et al. numbers.  Also, for a fair comparison of numbers in the Avg. column, especially to compare w/ Multilingual performance from Anwar et al, En results should not be included in Table 1.

* The model includes a prompt, but the value of this component is not clear or empirically determined.  Are the prompts sometimes there and other times not?  What is the performance if the prompts are not included in the model training/test?

* Paper lacks clarity / errors in places.  E.g.
     + 013-014: “ … both label and nuance”. Not clear what this means
     + 014-016: “ … predict correct speech with correct label” … is it predicting speech, or words?
     + 054-057: “ … fine-tuning followed by pre-training” or the other way around?
     + 219: Does Eq (12) need a sum over all samples?  Otherwise \gamma has no impact on the optimization.
     + Table 1 caption mentions boldfaced or underlined letters, but there are no such letters/numbers in the table.

**Reproducibility:**

3: Could reproduce the results with some difficulty. The settings of parameters are underspecified or subjectively determined; the training/evaluation data are not widely available.

**Reviewer Confidence:**

4: Quite sure. I tried to check the important points carefully. It's unlikely, though conceivable, that I missed something that should affect my ratings.

---

> ### Author Rebuttal · Authors · 2023-08-28
>
> We thank the reviewer for the appreciation of our paper and contribution. We hope this rebuttal helps to elucidate any confusing subjects such that reviewers would be open to improve their ratings. All feedback will be incorporated into our final manuscript.
>
> **(1) Fairness for performance comparison**
>
> We would like to thank the reviewer for the valuable concern. We would like to clarify that for the fair comparison, we have evaluated the recognition performances of all the languages using the pre-trained model provided by Anwar et al. We will mention about the re-evaluation of the previous method from the pre-trained model for the fair comparison for the future manuscript. Furthermore, to avoid the confusion and for a fair comparison, we would rewrite the performance comparison part of the manuscript and rearrange Table 1.
>
> **(2) Understanding of prompt**
>
> We design the novel language-representative prompt on top of the largely pre-trained audio-visual representation model in order to train the proposed architecture to not only determine the language of the input speech but also correctly predict what the input speech is delivering. The prompts are always trained in an end-to-end way along with the entire audio-visual multilingual speech recognition network architecture. Furthermore, we would like to clarify that the previous work (Anwar et al.,2023) has trained on the largely pre-trained audio-visual representation model (Shi et al., 2022), so we consider this as a baseline architecture that the reviewer has pointed out of excluding the language-representative prompt.
>
> **(3) Clarity of the manuscript**
>
> We would greatly appreciate that the reviewer points out the parts that are less clear to understand. To avoid confusion, we will reflect all the confusion in the future manuscript.
>
> - __*line 013-014: “ … both label and nuance”*__ &nbsp; Through the proposed prompt-finetuning technique using the largely pre-trained model, the single proposed model can determine which language the input speech is delivering (label), and it can also predict how the speech is delivering (nuance) for the correct prediction.
>
> - __*line 014-016: “ … predict correct speech with correct label”*__ &nbsp; We would like to clarify that the proposed network architecture is trained to predict a sentence (a set of words) from the input speech. Hence, from the high-level point of view, we mention that the proposed model predicts the correct speech.
>
> - __*line 054-057: “ … fine-tuning followed by pre-training”*__ &nbsp; We would like to thank the reviewer for the comment. We would make the clarification to “pre-training the multilingual speech recognition model or audio-visual speech representation model followed by fine-tuning the specific language”.
>
> - __*line 291: Eq (12)*__ &nbsp; We would like to clarify that the $\gamma$ is the distribution ratio for each language, so the $\gamma$ value changes every iteration regarding the distribution of the language in each mini-batch per iteration.
>
> - __*Caption of Table 1*__ &nbsp; We apologize for the misleading information in the caption of Table 1. We will change the caption of Table 1 to maintain the consistency in the future manuscript.

---

### Official Review · Reviewer_djfB · 2023-08-04

**Soundness:** 2

**Excitement:**

2: Mediocre: This paper makes marginal contributions (vs non-contemporaneous work), so I would rather not see it in the conference.

**Paper Topic And Main Contributions:**

The paper deals with multilingual audio-visual speech recognition. The authors describe a method that extracts a set of audio-visual features and augments them with  embeddings which have been trained to optimize a language classification loss. Experiments are evaluated on a dataset of multilingual video recordings of TED speeches.


**Reasons To Accept:**

The paper describes experiments in the area of multi-lingual multimodal speech recognition and a publishes a dataset that can be used by others for that purpose.

**Reasons To Reject:**

Generally, I don't agree with the conclusions that the authors derive from their experimental results. For instance, the authors claim an average 11% WER reduction due to the proposed method compared to their previous model. However, this compares results on two different test sets - the proposed model has been evaluated on English data, too, while the previous model has no result for English test data. If you exclude the English test data from the comparison (apple to apple) the previous model achieves an Avg WER of 39.51% on clean data and the proposed model achieves an Avg. WER of 39.45% which is not really an improvement. Furthermore, I believe the proposed approach misses a comparison to a multi-task training baseline which includes a language classification loss.

**Reproducibility:**

3: Could reproduce the results with some difficulty. The settings of parameters are underspecified or subjectively determined; the training/evaluation data are not widely available.

**Reviewer Confidence:**

2: Willing to defend my evaluation, but it is fairly likely that I missed some details, didn't understand some central points, or can't be sure about the novelty of the work.

---

> ### Author Rebuttal · Authors · 2023-08-28
>
> We thank the reviewer for the appreciation of our paper and contribution. We hope this rebuttal helps to elucidate any confusing subjects such that reviewers would be open to improve their ratings. We will incorporate all feedback in our final manuscript.
>
> **(1) Performance comparison**
>
> We would like to thank the reviewer’s valuable comments. We would like to point out that regarding the noisy environment, the proposed model achieved an average of 57.18 % WER excluding English, which shows a promising performance gain. This shows that the proposed model is also robust to the noisy environment, although it shows a small improvement in the clean environment. Regarding the reviewer’s concern, we would rewrite the claim of saying performance improvement and follow the fair comparison in the future manuscript to avoid the confusion.
>
> **(2) Comparison to multi-task baseline**
>
> We appreciate the reviewer’s suggestion. We would like to kindly clarify that we design a singly modeled multilingual audio-visual speech recognition technique with the use of a language classification module and the novel prompt finetuning technique. To the best of our knowledge, prior approaches in multilingual speech recognition related to multi-task learning have typically involved incorporating the language class as an input parameter, so they have conducted slightly different approach direction from the proposed model. We will add performance results from the additional previous technique that relates both the multi-task learning and the proposed architecture for the future manuscript.

---

### Meta-Review · Area_Chair_q9Rg · 2023-09-19

**Recommendation:** 4

**Metareview:**

The paper addresses multilingual audio-visual speech recognition and introduces a method for feature extraction and language classification loss optimization. Experiments are conducted on a multilingual dataset, MuAViC.

Pros:
- Innovative approach through fine-tuning with language prompts.
- Demonstrates robustness in low-resource languages and noisy environments.
- Addresses language imbalance with a weighted objective function.
- Comprehensive experiments on a challenging dataset.
- Potential impact on reducing language-specific models.

Cons:
- Issues with clarity, including unclear components and errors in the text.
- Inconsistent baseline numbers and unclear source for references.
- The value and impact of the prompt component are not adequately explained.
- Lack of an in-depth ablation study.
- No results for the scenario where the language is unknown during inference.
- Details about dataset language distribution are missing.
- Could explore the integration of recent advancements in multimodal learning.
- Focuses primarily on the MuAViC dataset; generalization across diverse data sources is not explored.


Overall, the paper's strengths lie in its innovative approach, robustness, and potential impact on multilingual speech recognition. However, there are areas of improvement related to clarity, experimental methodology, and further exploration of the proposed method's capabilities.

---

### Decision · Program_Chairs · 2023-10-07

**Decision:**

Accept-Findings

**Comment:**

The paper addresses multilingual audio-visual speech recognition and introduces a method for feature extraction and language classification loss optimization. Experiments are conducted on a multilingual dataset, MuAViC.

Pros:
- Innovative approach through fine-tuning with language prompts.
- Demonstrates robustness in low-resource languages and noisy environments.
- Addresses language imbalance with a weighted objective function.
- Comprehensive experiments on a challenging dataset.
- Potential impact on reducing language-specific models.

Cons:
- Issues with clarity, including unclear components and errors in the text.
- Inconsistent baseline numbers and unclear source for references.
- The value and impact of the prompt component are not adequately explained.
- Lack of an in-depth ablation study.
- No results for the scenario where the language is unknown during inference.
- Details about dataset language distribution are missing.
- Could explore the integration of recent advancements in multimodal learning.
- Focuses primarily on the MuAViC dataset; generalization across diverse data sources is not explored.


Overall, the paper's strengths lie in its innovative approach, robustness, and potential impact on multilingual speech recognition. However, there are areas of improvement related to clarity, experimental methodology, and further exploration of the proposed method's capabilities.